# Physicians’ Controversies Towards Fertility Preservation in Young Patients with Gynecological Cancer: An MITO Survey

**DOI:** 10.3390/curroncol32090527

**Published:** 2025-09-21

**Authors:** Giacomo Corrado, Inge Peters, Erica Silvestris, Raffaella Cioffi, Marcello Iacobelli, Emanuela Mancini, Riccardo Vizza, Sofia Thiella, Gennaro Cormio, Sandro Pignata, Giorgia Mangili

**Affiliations:** 1Gynecologic Oncology Unit, Department of Woman’s and Child Health and Public Health Sciences, Agostino Gemelli University Polyclinic, IRCCS, 00136 Rome, Italy; ingetheodoraanne.peters@guest.policlinicogemelli.it; 2Oncology Unit IRCSS Cancer Institute “Giovanni Paolo II”, 70124 Bari, Italy; e.silvestris@oncologico.bari.it (E.S.); g.cormio@oncologico.bari.it (G.C.); 3Department of Obstetrics and Gynecology, San Raffaele Scientific Institute,20132 Milan, Italy; cioffi.raffaella@hsr.it (R.C.); mangili.giorgia@hsr.it (G.M.); 4Gynecologic Oncology Unit and Oncofertility Center, IRCCS—Regina Elena National Cancer Institute, 00144 Rome, Italy; marcello.iacobelli@ifo.it (M.I.); emanuela.mancini@ifo.it (E.M.); 5Unit of Obstetrics and Gynecology, Department of Surgery, Dentistry, Pediatrics and Gynecology, AOUI Verona, University of Verona, 37126 Verona, Italy; riccardo.vizza@studenti.univr.it; 6Center for Research and Studies on Reproductive Health, Catholic University of the Sacred Heart, 00168 Rome, Italy; sofia.thiella01@icatt.it; 7Department of Interdisciplinary Medicine (DIM), University of Bari “Aldo Moro”, 70121 Bari, Italy; 8Uro-Gynecological Department, National Cancer Institute of Naples Fondazione G Pascale IRCCS, 80131 Naples, Italy; s.pignata@istitutotumori.na.it

**Keywords:** fertility preservation, controversies, gynecological cancer, survey

## Abstract

Due to the increasing incidence of malignant tumors among adolescents and young adults, as well as the growing demand for improved quality of life, physicians must know how to manage fertility preservation in oncological patients. Indeed, although guidelines have been developed, the management practices remain debatable for some tumors. The present survey highlights the controversies in fertility preservation for gynecological cancers and directs attention to knowledge gaps in these guidelines.

## 1. Introduction

The incidence rate of all cancers increased by 29% between 1973 and 2015 in adolescents and young adults of both sexes [1]. In Italy, every day, at least 30 new cancer cases are diagnosed in patients younger than 40 years of age, which equals 3% of new cancer diagnoses [2]. Interest in fertility preservation (FP) has increased in recent decades. Firstly, this is due to the fact that women are more often delay childbearing; in Italy, the percentage of pregnancies registered among women over the age of 35 years was a mere 12% in 1990 and is currently estimated to be 25% [3]. Secondly, anti-cancer therapies have significantly improved the survival of young cancer patients and, therefore, the possibility to resume life following cancer treatment [4].

Gynecological cancer directly affects fertility, as standard treatment consists of surgical removal of the reproductive organ and/or exposure to gonadotoxic treatment modalities. However, patients who present with early-stage gynecological cancer who meet strict criteria might be treated with fertility-sparing surgeries, thereby achieving oncological results equivalent to those undergoing traditional treatments [5]. Furthermore, FP techniques such as the cryopreservation of oocytes, embryos, and ovarian tissue might be offered in some situations [6,7]. Accordingly, the recent guidelines from the European Society of Gynaecological Oncology (ESGO), the European Society of Human Reproduction and Embryology (ESHRE), and the European Society for Gynaecological Endoscopy (ESGE) recommend FP counseling with a reproductive specialist who has an in-depth understanding of the patient and the couple’s history [8].

We decided to explore the current knowledge, attitudes, and clinical practices of physicians involved in the treatment of gynecological cancer towards the current controversies related to FP in young gynecological cancer patients.

## 2. Material and Methods

A specially developed questionnaire on fertility issues in gynecologic cancer patients was administered by email to 167 representatives of 167 centers of the Multicenter Italian Trials in Ovarian cancer and gynecologic malignancies (MITO) group. The MITO group represents the most important Italian scientific society, in which gynecologic oncologists, medical oncologists, radiation oncologists, or reproductive endocrinology and infertility (REI) physicians are involved in 167 centers across the country.

The survey was open for responses from 1 February 2023 to 30 July 2023. The email contained a link through which representatives could access the survey. The email address was not linked to the survey and no personal information was required. The entire process was anonymous.

### Characteristics of the Survey

The questionnaire was approved by the MITO internal review board. It consisted of 53 items and was divided into 4 main sections—(1) sociodemographic information; (2) knowledge and availability of FP procedures; (3) attitudes about FP controversies in gynecological cancers; and (4) attitudes about controversies on artificial reproductive technologies and genetic issues.

The data were analyzed using descriptive statistical methods, calculating the absolute and percentage frequencies (N, %) for all qualitative variables. Statistics were calculated using SPSS version 27 (IBM, Armonk, NY, USA).

## 3. Results

A total of 56 out of 167 MITO centers responded to our survey (response rate: 33.5%). Most respondents were women (64.3%). A total of 87.5% of respondents were Catholic and 53.6% of respondents were from northern Italy. The responding physicians were gynecologists (58.9%), followed by medical oncologists (33.9%), radiation oncologists (3.6%), and fertility specialists (3.6%). More than half of them (60.7%) worked in a Gynecological Oncology unit. Further details are displayed in Table 1.

The vast majority of respondents (91.1%) reported consulting current national and international guidelines on FP in patients with cancer and/or cancer survivors on a regular basis. Only three respondents (5.4%) declared not to be aware of the availability of guidelines on this topic.

### 3.1. Knowledge About Fertility Preservation Techniques

Approximately half of the physicians who participated in the survey stated that they had adequate knowledge about the use of gonadotropin-releasing analog (GnRHa) injections (n = 30; 53.6%), the cryopreservation of oocytes (n = 25; 44.6%), and the cryopreservation of ovarian tissue (n = 27; 48.2%). However, a total of 7 (12.5%), 10 (17.9%), and 14 (25.0%) respondents reported that they were aware of GnRHa, the cryopreservation of oocytes, and the cryopreservation of ovarian tissue, respectively, but did not consider themselves well-informed (Figure 1).

The availability of the administration of GnRHa, oocyte cryopreservation, and ovarian tissue cryopreservation within the respondents’ work settings was 92.9%, 67.9%, and 44.6%, respectively (Figure 2).

Furthermore, respondents were asked to indicate what suggestions might help to improve FP services in cancer patients. The following items were considered valuable or highly valuable: the increment of patient awareness (n = 49; 87.5%), FP training for professionals involved in the treatment of cancer patients (n = 54; 96.4%), the role of specialized nurses (n = 43; 76.8%), the standard offering of onco-fertility consultations with a gynecologist for all women affected by cancer at reproductive age (n = 49; 87.5%), and agreement between departments on who is responsible for discussing FP in young patients with cancer (n = 51; 91.1%). Further details are displayed in Table 2.

### 3.2. Artificial Reproduction Techniques

Overall, 52 respondents (92.9%) agreed on the statement that the cryopreservation and autotransplantation of ovarian tissue must be performed exclusively in highly specialized referral centers. Furthermore, 31 of the respondents (55.3%) declared that the cryopreservation of ovarian tissue must be performed only in patients who have not yet been exposed to potentially gonadotoxic chemotherapy regimens.

Contrasting opinions were found on the proposition that the cryopreservation of ovarian tissue should be limited to patients under the age of 35 years in order to increase the potential for the restoration of ovarian function at the time of the autotransplantation of frozen and thawed ovarian tissue fragments. Similarly, opinions were divided as to whether ovarian suppression with GnRH analogs should exclusively be offered to patients in whom the cryopreservation of oocytes or ovarian tissue is not considered feasible.

Physicians’ attitudes towards controversies over artificial reproductive technologies are shown in Table 3.

### 3.3. Fertility Preservation in Borderline Ovarian Tumors or Malignant Ovarian Tumors

Physicians’ attitudes towards controversies on FP in gynecological cancers are shown in Table 4.

A total of 38 respondents (67.8%) agreed on the option to perform controlled ovarian stimulation with subsequent oocyte cryopreservation in patients undergoing surgery for borderline ovarian tumors (BOTs). In contrast, only 19 (33.9%) of the respondents agreed on the proposition that the ovarian tissue cryopreservation of the normal-appearing contralateral ovary could be performed in patients undergoing unilateral adnexectomy because of BOTs. A total of 24 respondents (42.8%) were neutral or disagreed.

A total of 33 respondents (58.9%) agreed on the statement that oocyte cryopreservation should be offered to women diagnosed with malignant ovarian germ cell tumors.

Finally, 19 respondents (33.9%) would not consider FP in patients diagnosed with granulosa cell tumor with concomitant atypical endometrial hyperplasia.

### 3.4. Endometrial Cancer

For patients with low-risk endometrial cancer who have had a complete histological response after progesterone therapy and who have not yet spontaneously become pregnant six months after progesterone treatment, 45 respondents (80.3%) believed these patients should be offered artificial reproductive technologies (e.g., in vitro fertilization) to increase fecundity. Furthermore, 27 respondents (48.2%) felt it is opportune to consider reinitiating progesterone treatment in patients with recurrent endometrial cancer who had a complete response after initial progesterone treatment and who had not yet completed their childbearing wishes. Nevertheless, a total of 13 (23.2%) and 12 (21.4%) respondents disagreed or were neutral, respectively, in attempting conservative treatment again.

A substantial proportion of respondents (57.1%) stated that hysterectomy should always be performed after a fulfilled child wish in patients with a previous diagnosis of endometrial cancer, regardless of the histological response status following progesterone treatment.

Furthermore, a total of 28 respondents (50.0%) agreed that conservative management can be considered in patients with grade 2 endometrial cancers without myometrial invasion. Finally, 31 respondents (55.2%) expressed disagreement or neutrality to the statement that oral progesterone or an intrauterine progesterone device is equally effective for conservative management in patients with endometrial tumors.

### 3.5. Cervical Cancer

The majority of respondents (n = 36; 64.3%) felt that frozen sections on resected sentinel lymph nodes can be used as an intraoperative decision-making tool for FP in early-stage cervical cancers. In addition, 29 respondents (51.8%) agreed on administering neoadjuvant chemotherapy exclusively in patients participating in clinical trials with cervical tumors larger than 2 cm. Ovarian transposition and ovarian tissue autotransplantation were deemed safe in patients with cervical cancer with lymphovascular space invasion (LVSI) according to 27 respondents (48.2%). This was considerably higher as compared to those who remained neutral or expressed disagreement, at 26.8% and 21.4%, respectively.

A total of 49 respondents (87.5%) agreed or were neutral on considering oocyte cryopreservation and/or ovarian transposition in cervical cancer patients in whom adjuvant radiotherapy is recommended after hysterectomy. Moreover, 20 respondents (35.7%) also considered uterine transplantation in patients undergoing radical hysterectomy for early-stage cervical cancer, while 15 respondents (26.8%) disagreed.

### 3.6. Genetic Issues

Physicians’ attitudes towards controversies on genetic issues are shown in Table 5.

Approximately 35 respondents (62.5%) believed that patients younger than 40 years of age at the time of primary diagnosis of endometrial cancer should be referred to a clinical geneticist, regardless of mismatch repair status. Prescribing progesterone treatment in patients with atypical endometrial hyperplasia or endometrial cancer carrying genetic mutations associated with Lynch syndrome produced conflicting results. In fact, 20 survey participants (35.6%) did not agree with the use of these drugs, whereas 21 (37.5%) did not provide an answer.

Many respondents (n = 39, 69.7%) agreed to always offer fertility cryopreservation in BRCA gene mutation carriers as these patients may have a reduced ovarian reserve. A substantial proportion of respondents believed that preimplantation genetic testing should be discussed with patients who harbor pathogenic BRCA mutation variants (n = 42, 75%), those who are diagnosed with Lynch syndrome in whom endometrial aberrations have not yet occurred (n = 38, 67.8%), and those who are known to have genetic variants in homologous recombination genes other than BRCA1 or BRCA2 (e.g., RAD51C, RAD51D, BRIP1, etc.) (n = 36, 64.3%).

On the other hand, the respondents’ perspectives on offering pre-implantation genetic testing to patients carrying BRCA genetic variants of unknown significance (VUSs) were less consistent, as follows: 22 (39.8%) agreed, 13 (23.2%) remained neutral, and 21 (37.5%) disagreed. Regarding ovarian tissue autotransplantation, 51.8% (n = 29) did not agree on the proposition that this procedure is safe in carriers of *BRCA* genetic mutations, even if the ovarian tissue fragments would be re-implanted to the remaining ovary rather than the peritoneum. Conversely, 12 (21.4%) and 15 (26.8%) of the respondents expressed agreement or neutrality, respectively.

## 4. Discussion

This survey, conducted among MITO centers, highlighted the current landscape of awareness, attitudes, and practices in relation to FP among gynecologic oncology professionals in Italy. To our knowledge, this was the first survey to examine these aspects among physicians working in the gynecologic cancer field. Overall, our survey showed that physicians have divergent perspectives and approaches towards this topic. This is likely due to the relatively limited evidence available in the literature on FP in gynecologic cancer, which makes it difficult to provide unequivocal advice to gynecologic cancer patients who wish to preserve their fertility.

A notable finding was the high proportion of respondents (91.1%) who reported regularly consulting national and international guidelines on FP in cancer patients and survivors. This was consistent with previous studies indicating that standardized guidelines are critical to ensure best practice in oncofertility care [9].

### 4.1. Fertility Preservation Techniques

A considerable proportion of respondents expressed uncertainty about FP techniques, in accordance with previous studies highlighting gaps in FP education among oncology professionals [10]. Moreover, the availability of FP techniques was not equal between respondents’ workplaces; in particular, limited access to ovarian tissue cryopreservation reflects the need for increased infrastructure and training in this field [11].

Participants identified several key strategies to improve FP services, including increased patient awareness (87.5%) and FP training for professionals (96.4%). These findings were congruent with previous reports suggesting that educational interventions for both patients and providers improve FP decision-making [10]. Furthermore, agreement on defining responsibilities for discussing FP in young cancer patients (91.1%) demonstrated the importance of structured and multidisciplinary approaches in relation to oncofertility care [12,13].

Considering ovarian tissue cryopreservation, respondents had opposing opinions with respect to patient selection criteria [14]. For instance, 55.3% stated that ovarian tissue cryopreservation should be performed only in patients who had never undergone potentially gonadotoxic chemotherapy treatments. However, data are emerging in the literature that show the possibility of performing this treatment even after chemotherapy exposure [15,16,17].

### 4.2. Borderline Ovarian Tumors

Considering gynecological tumors, BOTs represent a unique challenge due to their uncertain malignant potential. While fertility-sparing surgery is an option for many women with BOTs [18], concerns about future fertility remain as ovarian cyst removal may lead to a reduced ovarian reserve, while the risk of recurrent disease cannot be ruled out. As a consequence, oocyte cryopreservation could be a valuable option for FP in individuals diagnosed with BOTs [19]. In our survey, the attitudes regarding ovarian tissue cryopreservation in BOT patients were conservative (33.9% agreement). This reflects ongoing ethical and clinical concerns about the safety and efficacy of FP interventions in specific tumor types [20].

### 4.3. Endometrial Cancer

For women with early, low-grade endometrial cancer who wish to preserve fertility, hysteroscopic endometrial resection followed by hormonal treatment is often considered a viable option [21]. In contrast, limited data are available on the impact of oocyte cryopreservation on oncological outcomes in women with endometrial cancer. Although ovarian stimulation did not seem to increase the risk of cancer recurrence or progression per se, more research is needed to determine the safety and efficacy of this approach in women with endometrial cancer [22,23,24]. Regarding secondary conservative treatment for recurrent endometrial cancer in patients who have had a complete response after initial diagnosis, 13 (23.2%) and 12 (21.4%) responding centers disagreed or were neutral, respectively, in relation to offering conservative treatment again. The ESGO/ESHRE/ESGE guidelines for the fertility-conserving treatment of patients with endometrial cancer indicate that a second conservative approach may be considered on a case-by-case basis, but the level of evidence remains low (level of evidence IV, grade B) [8].

### 4.4. Cervical Cancer

Treatment of cervical cancer may require radical hysterectomy or chemoradiation, compromising a woman’s ability to carry a pregnancy to term [25]. In these cases, surrogacy offers hope and provides the opportunity to have children despite cancer treatment [26]. Our survey shows that most physicians are positive regarding oocyte cryopreservation and/or ovarian transposition in relation to maintaining surrogacy as a viable option in cervical cancer patients who will undergo radical hysterectomy and/or chemoradiation. However, surrogacy in cervical cancer involves complex ethical and legal issues [27]. In patients in urgent need of anti-cancer treatments at risk of gonadotoxicity, ovarian tissue cryopreservation and autotransplantation have been proposed [28,29]. However, tumor cell reimplantation along with ovarian tissue transplantation cannot be excluded, particularly in non-squamous histology and advanced FIGO stages [30]. Furthermore, to date, no pregnancies have been reported following orthotopic transplantation of ovarian tissue fragments following pelvic radiotherapy [31].

### 4.5. Genetic Issues

About 30% of the respondents were not favorable to providing FP counseling to patients who carry a BRCA mutation. However, patients with BRCA mutations might have a higher risk of developing cancer and impaired fertility. For this reason, tailored oncofertility counseling should be provided to women at the time of BRCA mutation diagnosis [32]. Furthermore, significant uncertainty persists concerning the safety of preimplantation genetic testing and ovarian tissue autotransplantation in this population, underscoring the need for research on fertility interventions in individuals who have a genetic predisposition [33].

### 4.6. Limitations

The current study has several limitations. First, the response rate among MITO representatives was relatively low. Second, the degree of knowledge was self-reported and we did not test the respondents’ experience. Finally, although we believe that the questionnaire comprehensively addressed the current controversies regarding FP in gynecological cancer, it was not officially validated for this purpose.

In conclusion, our survey results demonstrated a wide variety of physicians’ perspectives, attitudes, and practices in relation to FP in young gynecological cancer patients. These findings emphasize the lack of evidence regarding many of these issues. In order to improve oncofertility counseling and adherence to available guidelines, training courses are urgently needed, as well as conducting research regarding the many gray areas in the field.

## Figures and Tables

**Figure 1 curroncol-32-00527-f001:**
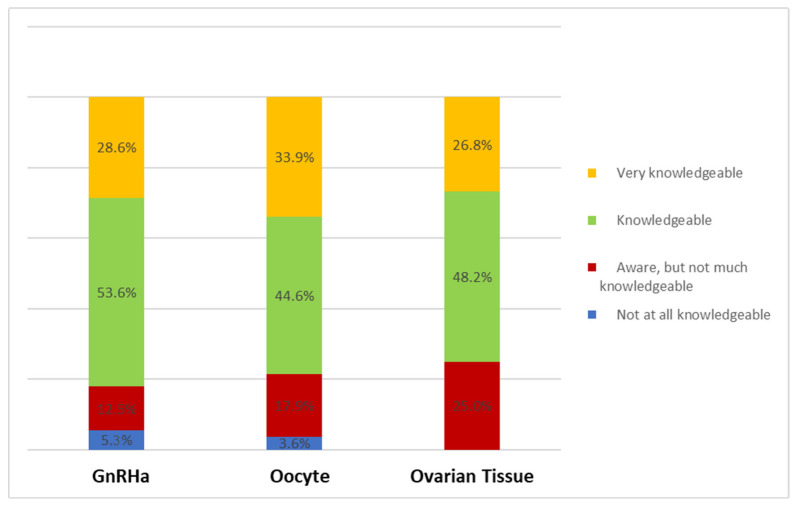
Knowledge of FP procedures as stated by respondents (N = 56).

**Figure 2 curroncol-32-00527-f002:**
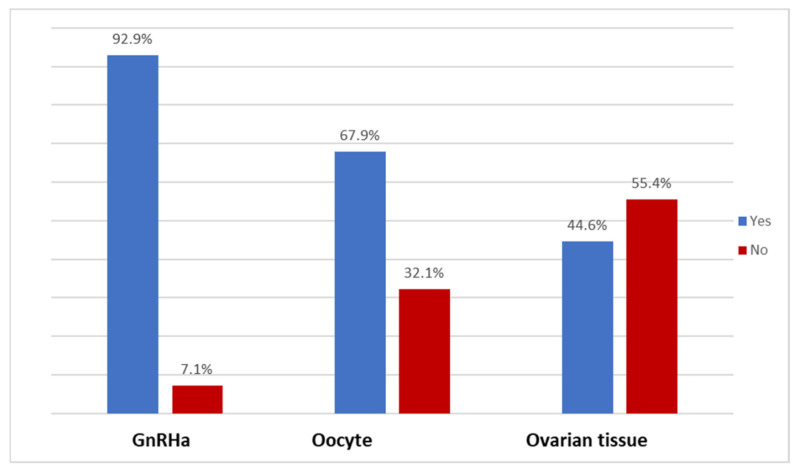
Availability of FP procedures for patients diagnosed with gynecological cancer at the 56 MITO centers (N = 56) wherein the respondents are employed.

**Table 1 curroncol-32-00527-t001:** Demographic information of physicians who responded to the survey.

Respondents	N = 56	%
Age (years)		
<40	21	37.5
40–50	16	28.6
>50	19	33.9
Gender		
Male	20	35.7
Female	36	64.3
Religion		
Catholic	49	87.5
Protestant	0	0.0
Muslim	0	0.0
Hindu	1	1.8
Jewish	0	0.0
Atheist/none	4	7.1
Prefer not to answer	2	3.6
Region of practice		
Northern Italy	30	53.6
Central Italy	12	21.4
Southern Italy	10	17.9
Italian islands (i.e., Sicily, Sardinia)	4	7.1
Specialty		
Gynecology	33	58.9
Medical oncology	19	33.9
Radiation oncology	2	3.6
Fertility specialist	2	3.6
Practice environment—1		
Specialized cancer center	18	32.1
Academic general hospital	20	35.7
Non-academic general hospital	17	30.4
Other	1	1.8
Practice environment—2		
Public	44	78.6
Private	5	8.9
Both	7	12.5
Working in a Gynecologic Oncology Unit		
No	22	39.3
Yes	34	60.7
Years of clinical experience		
<5 years	12	21.4
5–10 years	10	17.9
11–19 years	15	26.8
20–29 years	13	23.2
>30 years	6	10.7

**Table 2 curroncol-32-00527-t002:** Knowledge and availability of fertility preservation procedures in gynecological cancers.

Respondents	N = 56	%
Have you ever consulted some (inter)national guidelines on fertility preservation in patients with cancer and cancer survivors?		
No, I am not aware of available guidelines on this topic	3	5.4
No, but I know where to find these guidelines, if needed	2	3.6
Yes	51	91.1
How would you describe your knowledge of the use of GnRH analogs in patients with gynecological cancer?		
Not at all knowledgeable	3	5.4
Aware, but not very knowledgeable	7	12.5
Knowledgeable	30	53.6
Very knowledgeable	16	28.6
How would you describe your knowledge of oocyte cryopreservation in patients with gynecological cancer?		
Not at all knowledgeable	2	3.6
Aware, but not very knowledgeable	10	17.9
Knowledgeable	25	44.6
Very knowledgeable	19	33.9
How would you describe your knowledge of ovarian tissue cryopreservation in patients with gynecological cancer?		
Not at all knowledgeable	0	0.0
Aware, but not very knowledgeable	14	25.0
Knowledgeable	27	48.2
Very knowledgeable	15	26.8
Is ovarian suppression with GnRH analogs during chemotherapy available in your setting?		
No	4	7.1
Yes	52	92.9
Is oocyte cryopreservation available in your setting?		
No	18	32.1
Yes	38	67.9
Is ovarian tissue cryopreservation available in your setting?		
No	31	55.4
Yes	25	44.6
To what extent do you think each of the following suggestions that may improve female oncofertility care might be of value in your setting?		
Increment of patient awareness		
Useless	1	1.8
Not valuable	1	1.8
Neutral	5	8.9
Valuable	25	44.6
Highly valuable	24	42.9
Development and provision of patient information materials (e.g., decision aids, leaflets)		
Useless	2	3.6
Not valuable	7	12.5
Neutral	0	0.0
Valuable	27	48.2
Highly valuable	20	35.7
Education of professionals		
Useless	1	1.8
Not valuable	0	0.0
Neutral	1	1.8
Valuable	15	26.8
Highly valuable	39	69.6
Feedback to professionals on their performance		
Useless	1	1.8
Not valuable	0	0.0
Neutral	5	8.9
Valuable	29	51.8
Highly valuable	21	37.5
Role of specialized nurses		
Useless	1	1.8
Not valuable	2	3.6
Neutral	10	17.9
Valuable	21	37.5
Highly valuable	22	39.3
Fertility as a standard item at the multidisciplinary tumor board		
Useless	1	1.8
Not valuable	0	0.0
Neutral	2	3.6
Valuable	24	42.9
Highly valuable	29	51.8
Standard consultations with a gynecologist for all female cancer patients of reproductive age		
Useless	1	1.8
Not valuable	2	3.6
Neutral	4	7.1
Valuable	15	26.8
Highly valuable	34	60.7
Reminders in medical records		
Useless	1	1.8
Not valuable	0	0.0
Neutral	7	12.5
Valuable	24	42.9
Highly valuable	24	42.9
Agreement between healthcare departments on who is responsible for fertility discussions		
Useless	1	1.8
Not valuable	0	0.0
Neutral	4	7.1
Valuable	20	35.7
Highly valuable	31	55.4
Improved referral to fertility centers		
Useless	1	1.8
Not valuable	1	1.8
Neutral	2	3.6
Valuable	16	28.6
Highly valuable	36	64.3

**Table 3 curroncol-32-00527-t003:** Attitudes towards controversies on fertility preservation in gynecological cancers.

Respondents	N = 56	%
Borderline ovarian tumors and ovarian cancer		
Ovarian stimulation with subsequent oocyte cryopreservation could be considered in patients who underwent surgery because of borderline ovarian tumor		
Strongly disagree	0	0.0
Disagree	8	14.3
Neutral	10	17.9
Agree	27	48.2
Strongly agree	11	19.6
Ovarian tissue cryopreservation of the normally appearing contralateral ovary in patients who underwent unilateral surgery because of borderline ovarian tumor should be considered		
Strongly disagree	2	3.6
Disagree	12	21.4
Neutral	12	21.4
Agree	19	33.9
Strongly agree	11	19.6
Oocyte cryopreservation should be proposed to women diagnosed with malignant ovarian germ cell tumors		
Strongly disagree	2	3.6
Disagree	8	14.3
Neutral	13	23.2
Agree	27	48.2
Strongly agree	6	10.7
Patients diagnosed with granulosa cell tumors with concomitant atypical endometrial hyperplasia should be withheld from fertility preservation		
Strongly disagree	2	3.6
Disagree	21	37.5
Neutral	12	21.4
Agree	19	33.9
Strongly agree	2	3.6
Endometrial cancer		
Conservative management can be considered in patients with grade 2 endometrial cancers without myometrial invasion		
Strongly disagree	0	0.0
Disagree	15	26.8
Neutral	10	17.9
Agree	28	50.0
Strongly agree	3	5.4
Oral progesterone and progesterone intrauterine devices are equally effective as a form of conservative management in patients with endometrial cancers		
Strongly disagree	4	7.1
Disagree	18	32.1
Neutral	13	23.2
Agree	19	33.9
Strongly agree	2	3.6
Patients with low-risk endometrial cancer who had complete histological response following progesterone therapy and who have not yet become spontaneously		
pregnant six months after progesterone treatment should be offered artificial reproductive technologies (e.g., in vitro fertilization) to increase fecundity		
Strongly disagree	1	1.8
Disagree	2	3.6
Neutral	8	14.3
Agree	34	60.7
Strongly agree	11	19.6
Reinstatement of progesterone treatment should be considered in patients with recurrent endometrial cancer who had complete response		
following initial progesterone treatment and who have not yet fulfilled their childbearing wishes		
Strongly disagree	4	7.1
Disagree	13	23.2
Neutral	12	21.4
Agree	27	48.2
Strongly agree	0	0.0
Hysterectomy should always be performed following childbearing in patients with former endometrial cancer diagnosis, irrespective of histological response status		
Strongly disagree	1	1.8
Disagree	9	16.1
Neutral	5	8.9
Agree	32	57.1
Strongly agree	9	16.1
Cervical cancer		
Frozen sections of sentinel lymph nodes could be considered as an intra-operative decision tool for fertility preservation in early-stage cervical cancers		
Strongly disagree	1	1.8
Disagree	12	21.4
Neutral	7	12.5
Agree	31	55.4
Strongly agree	5	8.9
Neoadjuvant chemotherapy in patients with cervical cancers larger than 2 cm should be exclusively administered to those who participate in clinical trials		
Strongly disagree	0	0.0
Disagree	12	21.4
Neutral	5	8.9
Agree	29	51.8
Strongly agree	10	17.9
Ovarian transposition and ovarian tissue autotransplantation should not be considered safe in patients with cervical adenocarcinoma		
Strongly disagree	1	1.8
Disagree	18	32.1
Neutral	12	21.4
Agree	23	41.1
Strongly agree	2	3.6
Ovarian transposition and ovarian tissue autotransplantation should be considered safe in patients with early-stage cervical cancer		
showing lymph-vascular space invasion (LVSI)		
Strongly disagree	0	0.0
Disagree	12	21.4
Neutral	15	26.8
Agree	27	48.2
Strongly agree	2	3.6
Oocyte cryopreservation and/or ovarian transposition should be considered in cervical cancer patients in whom adjuvant radiotherapy following		
hysterectomy is advised based on final pathology results to preserve the option of genetic motherhood using a surrogate mother		
Strongly disagree	1	1.8
Disagree	6	10.7
Neutral	18	32.1
Agree	26	46.4
Strongly agree	5	8.9
Uterine transplantation should be further explored as an option to preserve motherhood in patients who underwent radical hysterectomy		
because of early-stage cervical cancer		
Strongly disagree	1	1.8
Disagree	15	26.8
Neutral	20	35.7
Agree	16	28.6
Strongly agree	4	7.1

**Table 4 curroncol-32-00527-t004:** Attitudes towards controversies on artificial reproductive technologies and genetics.

Respondents	N = 56	%
Artificial reproductive technologies		
Ovarian tissue cryopreservation and autotransplantation should exclusively be performed in highly specialized referral centers		
Strongly disagree	1	1.8
Disagree	3	5.4
Neutral	0	0.0
Agree	17	30.4
Strongly agree	35	62.5
Ovarian tissue cryopreservation should only be performed in patients who have not yet been exposed to potentially gonadotoxic chemotherapeutic regimens		
Strongly disagree	5	8.9
Disagree	9	16.1
Neutral	11	19.6
Agree	25	44.6
Strongly agree	6	10.7
Ovarian tissue cryopreservation should be limited to patients younger than 35 years of age to increase the potential for ovarian function restoration		
upon autotransplantation of freeze–thawed ovarian tissue fragments		
Strongly disagree	3	5.4
Disagree	12	21.4
Neutral	10	17.9
Agree	27	48.2
Strongly agree	4	7.1
Ovarian suppression with GnRH analogs should only be offered to patients in whom the cryopreservation of either oocytes or ovarian tissue is not deemed feasible		
Strongly disagree	6	10.7
Disagree	14	25.0
Neutral	11	19.6
Agree	20	35.7
Strongly agree	5	8.9
Genetic screening and genetic mutation carriers		
Patients younger than 40 years of age at the time of primary diagnosis of endometrial cancer should be referred to a clinical geneticist,		
irrespective of mismatch repair status		
Strongly disagree	1	1.8
Disagree	12	21.4
Neutral	8	14.3
Agree	23	41.1
Strongly agree	12	21.4
In patients with endometrial atypical hyperplasia or endometrial cancer who carry gene mutations associated with Lynch syndrome,		
progesterone treatment should not be prescribed		
Strongly disagree	2	3.6
Disagree	18	32.1
Neutral	21	37.5
Agree	14	25.0
Strongly agree	1	1.8
Since BRCA genetic mutation carriers may have a diminished ovarian reserve, fertility preservation should always be offered		
Strongly disagree	0	0.0
Disagree	4	7.1
Neutral	13	23.2
Agree	29	51.8
Strongly agree	10	17.9
Preimplantation genetic testing should be discussed with patients who carry pathogenic BRCA mutation variants		
Strongly disagree	0	0.0
Disagree	7	12.5
Neutral	7	12.5
Agree	31	55.4
Strongly agree	11	19.6
Preimplantation genetic testing should be discussed with Lynch syndrome carriers in whom endometrial aberrations have not yet occurred		
Strongly disagree	0	0.0
Disagree	9	16.1
Neutral	9	16.1
Agree	33	58.9
Strongly agree	5	8.9
Preimplantation genetic testing should be discussed with patients who carry genetic variants in homologous recombination genes		
other than BRCA1 or BRCA2 (e.g., RAD51C, RAD51D, BRIP1, etc.)		
Strongly disagree	0	0.0
Disagree	9	16.1
Neutral	11	19.6
Agree	28	50.0
Strongly agree	8	14.3
Preimplantation genetic testing should be discussed with patients who carry BRCA genetic variants of unknown significance (VUSs)		
Strongly disagree	5	8.9
Disagree	16	28.6
Neutral	13	23.2
Agree	17	30.4
Strongly agree	5	8.9
Ovarian tissue transplantation can be considered safe in BRCA genetic mutation carriers, as long as ovarian transplants are placed back		
in the remaining ovary rather than the peritoneum		
Strongly disagree	6	10.7
Disagree	23	41.1
Neutral	15	26.8
Agree	12	21.4
Strongly agree	0	0.0

**Table 5 curroncol-32-00527-t005:** Physicians’ attitudes towards controversies on genetic issues.

Respondents	N = 56	%
Artificial reproductive technologies		
Ovarian tissue cryopreservation and autotransplantation should exclusively be performed in highly specialized referral centers		
Strongly disagree	1	1.8
Disagree	3	5.4
Neutral	0	0.0
Agree	17	30.4
Strongly agree	35	62.5
Ovarian tissue cryopreservation should only be performed in patients who have not yet been exposed to potentially gonadotoxic chemotherapeutic regimens		
Strongly disagree	5	8.9
Disagree	9	16.1
Neutral	11	19.6
Agree	25	44.6
Strongly agree	6	10.7
Ovarian tissue cryopreservation should be limited to patients younger than 35 years of age to increase the potential for ovarian function restoration upon autotransplantation of freeze–thawed ovarian tissue fragments		
Strongly disagree	3	5.4
Disagree	12	21.4
Neutral	10	17.9
Agree	27	48.2
Strongly agree	4	7.1
Ovarian suppression with GnRH analogs should only be offered to patients in whom cryopreservation of either oocytes or ovarian tissue is not deemed feasible		
Strongly disagree	6	10.7
Disagree	14	25.0
Neutral	11	19.6
Agree	20	35.7
Strongly agree	5	8.9
Genetic screening and genetic mutation carriers		
Patients younger than 40 years of age at the time of primary diagnosis of endometrial cancer should be referred to a clinical geneticist, irrespective of mismatch repair status		
Strongly disagree	1	1.8
Disagree	12	21.4
Neutral	8	14..3
Agree	23	41.1
Strongly agree	12	21.4
In patients with endometrial atypical hyperplasia or endometrial cancer who carry gene mutations associated with Lynch syndrome, progesterone treatment should not be prescribed		
Strongly disagree	2	3.6
Disagree	18	32.1
Neutral	21	37.5
Agree	14	25.0
Strongly agree	1	1.8
Since BRCA genetic mutation carriers may have a diminished ovarian reserve, fertility preservation should always be offered		
Strongly disagree	0	0.0
Disagree	4	7.1
Neutral	13	23.2
Agree	29	51.8
Strongly agree	10	17.9
Preimplantation genetic testing should be discussed with patents who carry pathogenic *BRCA* mutation variants		
Strongly disagree	0	0.0
Disagree	7	12.5
Neutral	7	12.5
Agree	31	55.4
Strongly agree	11	19.6
Preimplantation genetic testing should be discussed with Lynch syndrome carriers in whom endometrial aberrations have not yet occurred		
Strongly disagree	0	0.0
Disagree	9	16.1
Neutral	9	16.1
Agree	33	58.9
Strongly agree	5	8.9
Preimplantation genetic testing should be discussed with patients who carry genetic variants in homologous recombination genes other than BRCA1 or BRCA2 (e.g., RAD51C, RAD51D, BRIP1, etc.)		
Strongly disagree	C	0.0
Disagree	9	16.1
Neutral	11	19.6
Agree	23	50.0
Strongly agree		14.3
Preimplantation genetic testing should be discussed with patients who carry *BRCA* genetic variants of unknown significance (VUSs)		
Strongly disagree	5	8.9
Disagree	16	28.6
Neutral	13	23.2.
Agree	17	30.4
Strongly agree	5	8.9
Ovarian tissue transplantation can be considered safe in *BRCA* genetic mutation carriers, as long as ovarian transplants are placed back the remaining ovary rather than the peritoneum		
Strongly disagree	6	10.7
Disagree	23	41.1
Neutral	15	26.8
Agree	12	21.4
Strongly agree	0	0.0

## Data Availability

The original contributions presented in this study are included in the article. Further inquiries can be directed to the corresponding author.

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
