# Peer review of "Physicians’ Controversies Towards Fertility Preservation in Young Patients with Gynecological Cancer: An MITO Survey"

_curroncol, 2025, doi:10.3390/curroncol32090527_

Round 1

Reviewer 1 Report

Comments and Suggestions for Authors

I appreciate the opportunity to review this manuscript aimed at exploring the current knowledge, attitudes, and clinical practices of physicians involved in the treatment of gynecological cancer regarding the current controversies related to fertility preservation in young gynecological cancer patients. The article is well-written, and the methodology is sound, with the authors noting the main weakness of the study, which is the low response rate received, which they incorrectly describe as "relatively low" when it is, in fact, "truly low."

In my opinion, the following should be corrected:

Introduction: In the second paragraph, reference 5 should be placed at the end of the sentence.

Materials and Methods: The process of anonymization should be explained.

Results: Table 5 contains typographical errors that make it unreadable. A better design for all tables would be advisable.

Author Response

1. Summary

Thank you very much for taking the time to review this manuscript. Please find the detailed responses below and the corresponding revisions in the re-submitted files.

2. Questions for General Evaluation

Reviewer’s Evaluation

Response and Revisions

Does the introduction provide sufficient background and include all relevant references?

Yes

/

Are all the cited references relevant to the research?

Yes

Is the research design appropriate?

Yes

Are the methods adequately described?

Can be improved

Are the results clearly presented?

Yes

Are the conclusions supported by the results?

Yes

3. Point-by-point response to Comments and Suggestions for Authors

Comments 1: Introduction: In the second paragraph, reference 5 should be placed at the end of the sentence.

Response 1: Thank you for pointing this out. We agree with this comment. Therefore, we have placed reference 5 at the end of the sentence. Page 2, line 62.

Comments 2: Materials and Methods: The process of anonymization should be explained.

Response 2: Thank you for pointing this out. We have, accordingly, modified the sentence to emphasize this point. This change can be found in page 2, lines 86-89.

‘The email contained a link through which representatives could access the survey. The email address was not linked to the survey, and no personal information was required. The entire process was anonymous.’

Comments 3: Results: Table 5 contains typographical errors that make it unreadable. A better design for all tables would be advisable.

Resonse 3: Thank you for pointing this out. We have, accordingly, reviewed all tables and the English has been checked by a native speaker.

4. Response to Comments on the Quality of English Language

Point 1: The English is fine and does not require any improvement.

Response 1: Thank you.

Reviewer 2 Report

Comments and Suggestions for Authors

Dear Authors,

Thank you for the opportunity to review your manuscript. I sincerely appreciate the time and effort you have invested in this paper. The authors present a survey conducted by the MITO group on fertility preservation in gynecological cancer.

I agree that the topic is both relevant and timely; however, the study is limited by the low number of responses (56/790), and 39.3% of respondents do not work in an oncology unit, which restricts the strength of the conclusions.

Additionally, one of the survey questions addressed religion and how beliefs might influence indications for fertility preservation. I suggest that the authors provide a discussion on this point, as it could offer important context for interpreting the results.

Author Response

1. Summary

2. Questions for General Evaluation

Reviewer’s Evaluation

Response and Revisions

Does the introduction provide sufficient background and include all relevant references?

Yes

Are all the cited references relevant to the research?

Yes

Is the research design appropriate?

Yes

Are the methods adequately described?

Can be improved

Are the results clearly presented?

Can be improved

Are the conclusions supported by the results?

Can be improved

3. Point-by-point response to Comments and Suggestions for Authors

Comment 1: I agree that the topic is relevant and timely; however, the study is limited by the low response rate (56/790), and 39.3% of respondents do not work in an oncology department, which limits the validity of the conclusions.

Response 1: Thank you for pointing this out. We sincerely apologize for not being clear enough in the manuscript regarding the low response rate. MITO has 790 members. We emailed the survey to 167 members representing 167 MITO centers. The response rate was 56/167 (33.5% response rate). To avoid further misunderstanding, we have edited the manuscript. The changes can be found on page 1, line 42; page 2, line 81; Page 16, line 292.

Regarding the fact that 39.3% of respondents do not work in an oncology department, we would like to clarify that not all centers in our country have a Gynecologic Oncology department or an Oncofertility Unit. For this reason, 39.3% responded this way, primarily because they are part of a general Obstetrics and Gynecology department. This department provides medical care to cancer patients with dedicated medical staff and participates in the decisions of the Medical Tumor Board.

Comment 2: Additionally, one of the survey questions addressed religion and how religious beliefs may influence indications for fertility preservation. I suggest the authors provide a discussion on this point, as it could provide important context for interpreting the results.

Response 2: Thank you for pointing this out. It is correct that one of the survey questions addressed religion (Table 1), however, we did not ask respondents how religious beliefs may influence indications for fertility preservation. In our country, Italy, the majority of people are Catholic (87.5% in our survey), but religion does NOT influence medical decision-making in any way. Therefore, we do not consider it necessary to discuss this point in our manuscript.

4. Response to comments on the quality of the English language

Point 1: The English is good and does not require improvement.